# Natural Products Are a Promising Source for Anthelmintic Drug Discovery

**DOI:** 10.3390/biom11101457

**Published:** 2021-10-04

**Authors:** K. L. T. Dilrukshi Jayawardene, Enzo A. Palombo, Peter R. Boag

**Affiliations:** 1Development and Stem Cells Program, Monash Biomedicine Discovery Institute, Monash University, Melbourne, VIC 3800, Australia; diluvet@gmail.com; 2Development and Stem Cells Program, Department of Biochemistry and Molecular Biology, Monash University, Melbourne, VIC 3800, Australia; 3Department of Chemistry and Biotechnology, School of Science, Computing and Engineering Technologies, Swinburne University of Technology, Melbourne, VIC 3122, Australia

**Keywords:** anthelmintic, traditional medicine, drug-resistance, natural products, drug discovery

## Abstract

Parasitic nematodes infect almost all forms of life. In the human context, parasites are one of the major causative factors for physical and intellectual growth retardation in the developing world. In the agricultural setting, parasites have a great economic impact through a reduction in livestock performance or control cost. The main method of controlling these devastating conditions is the use of anthelmintic drugs. Unfortunately, there are only a few anthelmintic drug classes available in the market and significant resistance has developed in most of the parasitic species of livestock. Therefore, development of new anthelmintics with different modes of action is critical for sustainable parasitic control in the future. The drug development pipeline is broadly limited to two types of molecules, namely synthetic compounds and natural plant products. Compared to synthetic compounds, natural products are highly diverse, and many have historically proven valuable in folk medicine to treat various gastrointestinal ailments. This review focus on the use of traditional knowledge-based plant extracts in the development of new therapeutic leads, the approaches used as screening techniques, and common bottlenecks and opportunities in plant-based anthelmintic drug discovery.

## 1. Introduction

Parasites depend on their hosts for physiological and metabolic needs during the infective stages. Most host-parasite relationships have endured for thousands of years and often display remarkable host-parasite specificity. In this parasite-host co-evolutionary arms race, parasites have the advantage of short generation times and generally large number of progeny compared to their host species. As a result, many parasites can overcome the host resistance strategies [1] and treatment with anti-parasitic drugs is required to overcome the infection. The global distribution of nematode parasites results in a significant impact on human health and agricultural outputs; however, this disproportionately affects developing countries. The focus of this review is to compare drug discovery strategies over the last 20 years to drive developments of anti-helminth agents based on traditional knowledge of anthelmintic treatments.

### 1.1. Parasitic Infections in Humans

Despite improved healthcare systems around the globe over recent decades, parasitic infections are still strikingly common among humans [2]. It is estimated that approximately every second person in the world is affected by a parasitic infection [3]. Many common infectious diseases in humans are caused by soil-transmitted helminths which are a major cause of intellectual and physical growth retardation in developing countries [4,5]. Helminths are divided into several phyla, with the nematodes (roundworms) the most common with an estimated 800,000 to 1,000,000 species, of which over 300 are known to infect humans [6]. However, this is likely an underestimate due to lack of quality data and that the severe forms of the worm infection occur in only a few people and therefore lead to underreporting of cases [7]. However, parasitic nematodes lead to substantial long term impact on adult human health and considerable suffering in children (Table 1) [8].

Helminths infect humans when they contact parasitic eggs or larvae, which thrive in warm and moist soil [17]. Despite the enormous scale of the infection burden, these diseases are considered “Neglected Tropical Diseases” (NTDs) of humans as they are more prevalent in developing countries and are associated with low-income groups [18,19,20]. By contrast, parasitic infections in humans are relatively uncommon in developed regions and are conceptually considered acute health problems that usually lead to minimal morbidity [4]. However, helminth infections exacerbate the symptoms of major infectious diseases such as tuberculosis, HIV/AIDS and malaria [21,22] and make the host vulnerable to other opportunistic pathogens, like yeasts [23]. Therefore, irrespective of where they occur, parasitic helminths are still considered one of the leading causes of major human health problems and require sustained control efforts.

### 1.2. Parasitic Infections in Animals

Similar to the human experience, animals are also impacted by parasites that result in significant economic loss. Livestock production systems are important capital assets that provide more than half of the global agricultural production [24,25]. Parasitic diseases of agricultural animals have a broad global distribution and cause serious economic losses in both developed and developing countries. Among the parasitic infections, those resulting from gastrointestinal nematodes (GIN) are a leading cause of production loss in agricultural animal systems worldwide [26]. European-wide analysis of the economic burden of selected gastrointestinal nematodes revealed an annual cost of €1.8 billion. Eighty-one percent of this cost was due to loss of production and rest is due to treatment costs [27] (Figure 1). The effects of parasitic helminths in agricultural animals are different in developed and developing countries. In developed countries, the greatest impact is the cost of control of parasitic diseases, whereas, in developing countries, the impact is principally due to loss of productivity [28]. Overall, GINs can lead to decreases in growth rates, fertility, milk production, and meat and wool quality [29,30]. In addition to decreased production, GINs can lead to high, but variable, rates of mortality among herds depending on the causative agent and the age group of affected host individuals [31]. Unfortunately, detailed worldwide data on the cost of helminth infection are lacking [32] and current evaluations likely underestimate of the scale of the problem.

## 2. Anthelminthics

There are two general strategies to control parasitic infection among animals. One is the application of good animal husbandry or hygienic practices in order to minimise the introduction and spread of parasitic infections and the second is the use of chemicals/drugs to clear the infection. In the 18th century, chemical treatments based on the use of crude plant extracts or metals that had the ability to “irritate” the parasitic worm and/or remove them mechanically from the gut were introduced [33]. Some of the agents used include arsenic, common salt, and turpentine oil, and would likely have had a broad pleiotropic impact on the host and parasite. A new paradigm was established in the 1940s with the introduction of molecules that were both chemically defined and target specific [33]. The first of these was phenothiazine, an organic compound that was used to treat sheep parasitic diseases [34] and was followed by the development of additional drugs with greater efficacy. However, the relatively poor commercial returns have resulted in only a modest number of new anthelmintics being commercially released in the last two decades. Anthelmintics have revolutionized human, agricultural and companion animal health and the extraordinary success of these drugs provided significant global improvements in health and economic development [35]. Currently, there are a few main classes of anthelmintics: benzimidazoles (BZ), imidazothiazoles, tetrahydropyrimidines, macrocyclic lactones (ML), amino-acetonitrile derivatives (AAD), salicylanilides and organophosphates [36,37,38,39]. The precise modes of action of many anthelmintics are not known yet, although the site of their action and biochemical mechanism that has been altered in parasites upon their administration have been explored for many anthelmintics. The main target sites of the currently available anthelmintics are solely proteins and those are ion channels, microtubules and bioenergetics (Table 2) [40]. All the commercial anthelmintics included in Table 2 are non-plant derived chemicals and their activity is based on a single target species [41]. Most of these classes of drugs act on Cys-loop ligand-gated ion channels (LGICs) in invertebrates [42] and many were initially identified through low throughput approaches using either whole animal-based or phenotypic screening methods. However, some of the currently used anthelminthics showed detrimental effects on the normal microbiota [43] which increases the need for new anthelmintic compounds with minimal host toxicity and environmental impacts.

### Anthelmintic Resistance

Worryingly, similar to antibiotic resistance in bacteria, resistance to all the available anthelmintic drug classes has been reported in livestock species. What surprised many is the rate at which resistance has developed, often within a few years of introduction (Figure 2), which likely reflects the combination of large genetically diverse parasite populations and the strong selection pressure for resistance. Importantly, the time to resistance has been shorter than the cycle of new anthelmintic development. Although less prevalent, drug resistance has been documented for some of the most commonly used anthelmintics (mebendazole and pyrantel) against human hookworms [52,53]. There is also evidence for reduced drug efficacy for *Onchocerca vulgaris* [54,55] and for praziquantel-resistance in schistosomes. Interestingly, praziquantel resistance appears to be the result of mutations in the gene encoding sulfotransferase, the drug-activating enzyme and not the activated drug target [56,57].

Resistance to anthelmintics is a complex problem because the pathogens are a heterogeneous group with a complex life cycle. Different helminth species have distinct life cycles and different stages that rely on the host. For example, in humans, the ingestion of *Trichuris* eggs leads directly to gut infections, while the ingestion of *Ascaris* eggs and *Strongyloides* larvae leads to a pulmonary migration phase before gut infection. A critical element in the development of resistance is the fact that all the available anthelmintics target individual proteins that are encoded by one gene. A single mutation in that gene can lead to modified drug binding and increase resistance, and the strong selective pressure for the resistance mutation (s) can quickly spread within parasite populations [63]. For example, multiple independent monepantel-resistant parasitic worms have been identified in New Zealand (*Teladorsagia circumcincta* and *Trichostrongylus colubriformis*) [64] and in *Haemonchus contortus* populations in the Netherlands and Australia [65,66]. Monepantel acts on the nicotine acetylcholine receptors of nematodes and results in paralysis. Analysis of monepantel-resistant *Haemonchus contortus* populations showed they have several mutations in the target gene (*Hco-mptl-1*). These mutations were predicted to result in truncated proteins that were no longer effective monepantel binding targets [67] (Table 3). Importantly, the presence of multiple independent mutations in the *Hco-mptl-1* gene within this population of resistant worms suggests these mutations provide significant advantage and quickly establish within wild populations. Given the increasing prevalence of resistance, there is an urgent need to shorten the time required to developed new anthelmintics that will reduce the rate of resistance developing.

## 3. Plant Extracts as a Source of New Anthelmintics Compounds

There is a long history of using plant extracts to treat human and animal diseases. For example, the use of poppy (*Papaver somnniferum*) as an analgesic agent was described 4000 years ago [76]. Remarkably, the World Health Organization (WHO) found that as much as two-thirds of the world’s population depend on plants as primary agents to resolve health issues [77]. Before the 19th century, medicinal plants were used for various ailments based on experience without any mechanistic knowledge of their usage. Rational drug discovery started in the 19th century with the isolation of the analgesic morphine [78]. Not surprisingly, 50,000 to 70,000 plant species are used in both traditional and Western medicine approaches [79,80] and nearly one-quarter of prescription medicines are derived from plants or plant-derived secondary metabolites [81]. Plants synthesize a wide array of compounds which serve specific roles in metabolism. Natural compounds obtained from plant extracts can be chemically highly diverse when compared to synthetic compounds and often lead to specific biological activities [82]. With respect to parasitism, medicinal plants have been used for centuries to combat these diseases and are still used for this purpose in many parts of the world [83]. A growing area of investigation is the identification of new anthelmintics from plant extracts. Several extracts from plants have shown potential for development as anthelmintic agents [84,85,86,87,88,89] and the screening attempts and scientific validation of the number of products with anthelmintic potential has continued over the past twenty years (Figure 3). This increased attention on plant extracts for the identification of new anthelmintics is likely to significantly increase in the future as anthelmintic resistance to current drugs increases. Between 2000 to 2019, 40 patents were issued for natural products-based nematocides which includes seven structural classes [90], although none are yet commercially available. The majority of the natural product anthelmintics in the market and those patented since 2000 are derivatives of microbial products [90]. However, to fully realise the potential of plant extracts, the significant challenge of identifying the active component of complex chemical mixtures needs to be overcome.

### Screening Techniques of Plant-Based Compounds

Anthelmintic drug discovery has largely been based on three distinct screening approaches: animal-based screening, target-based screening and nematode phenotypic screening (whole organism-based assay) (Figure 4) [91,92]. All the available anthelmintic drug classes (at least prototypes of the drug classes) were found through low-throughput screening methods like targeting parasitic species in the host animal or in vitro assays [42,93]. Animal-based screening is the oldest method of anthelmintic drug discovery and is based on exposing infected animals to test compounds and then monitoring parasite fecundity, survival, and host toxicity. Natural products-based compounds that show some efficacy, often referred to as lead compounds, can be chemically modified to try and enhance the specificity and efficacy, and reduce host toxicity. Test compounds are mainly “new” synthetic or natural compounds; however drug repurposing is also an alternative approach in anthelmintic discovery [94]. The advantage of repurposing existing drugs can be the substantial reduction in time and cost of getting the drug to market [95,96].

Target-based and mechanism-based drug screening requires the identification of a specific target predicted to have a major impact on the development or function of the parasite. Once a target is identified, screening for new inhibitors or antagonistic compounds that impact target function follows and is accelerated by High Throughput Screening (HTS) methods. Generally, in these screens, chemical libraries are tested on known or predicted parasitic targets for possible interaction [97,98]. This method facilitates the identification of the interaction of tested compounds with the respective target, however, requires significant investment in sophisticated infrastructure and ongoing cost of production of the target proteins. Moreover, the low rate of “hits” that are active in the living system once evaluated after in vitro screening may limit the use of these techniques [92].

Whole organism-based approaches are used to detect potential compounds that initiate paralysis, abnormalities in feeding and reproduction or death in the target species [92]. Phenotypic screening relies on identifying substances such as small molecules, peptides, or chemicals that have a biological activity that alter the phenotype of a cell or an organism to combat the stress induced by the new drug candidate [42,94,99]. The recent development of high throughput whole organism assays has allowed rapid screening of chemical compounds for anthelmintic activity against different parasitic stages in *Haemonchus contortus* [100]. Despite these advances, drug development is limited since most of the parasitic stages of helminths cannot be maintained in vitro and some models fail to show the same susceptibility and bioaccumulation of anthelmintic compounds as parasitic species [99].

Another popular strategy to search for novel molecules is based on evidence through ethnopharmacology. Ethnopharmacology is scientific validation of the uses of traditional or “folk” medicine including medicinal plants used by the ancient healers. Traditionally, plants have been used for the treatment of helminth parasites as a home remedy in the form of crude preparations or after random testing on parasitic problems of already prepared herbal medicines for other diseases [101,102,103]. We can now investigate these traditional medicines via modern scientific practices to determine the pharmacological activity and efficacy using standard experimental models [104]. Once the activity of the crude plant extract is shown, the next step is the identification of the active ingredient through bio-guided extraction using forward pharmacological or phenotypic drug discovery approaches (Figure 5). In bio-guided extraction, plant materials are extracted using various solvents and tested on a biological model until the most active fraction (s) is identified using various methods of fractionation such as automated flash chromatography (FC), batch-throughput solid-phase extraction (SPE), high-performance liquid chromatography (HPLC) and high-throughput parallel eight-channel analytical liquid chromatography-evaporative light scattering detection-mass spectrometry (LC–ELSD–MS) [105]. The composition and structure elucidation can then be determined using modern pharmacognosy and chemical analytical methods such as mass spectrometry and nuclear magnetic resonance (NMR) [106,107]. These active molecules can be tested on specific parasitic proteins that are predicted or known as drug targets such as enzymes in key biological pathways and receptors for ion channels in high throughput mechanism-based screening methods [78,98,100]. The receptor binding assay enhances the availability of mechanism-based drug discovery but, with reference to anthelmintic drug discovery, selection of invertebrate drug targets is difficult given the gaps we still have in understanding the biology of the parasites, especially once in the host. Further, though there are many drug targets suggested, only a few are validated for use in drug discovery [97]. Traditional medicines from plants have activity on different disease conditions [108], therefore bio-guided extraction can be performed on respective experimental models which can result in the identification of different molecules with a different mode of actions. Some traditional medicines may work via synergistic activities of distinct molecules within the extract and, in these cases, identification and isolation of the key components is problematic [109]. Importantly, the use of plant extracts containing synergistic activities of different molecules may help reduce the rapid development of resistance as the chance of developing simultaneous mutations in multiple genes will be significantly lower. Identification of the active molecules remains a major challenge in realising the potential of plant extract as a major new resource for drug discovery.

An alternative strategy for new anthelminthic identification is the reverse pharmacological approach which involves target-based drug screening techniques. In this approach, plants are examined, and libraries of molecules are established. The molecules are tested on putative targets, i.e., enzymes or receptors involved in important biological activities in a cell. These chemical libraries are either based on random plants (unbiased compound libraries) or traditional use/knowledge-based libraries [78]. Some molecules might also act on as yet undefined targets and so would not be identified when screened for activity against putative “druggable” targets such as enzymes, receptors and ion channels and key molecules in cell signalling and biochemical pathways [110]. The reverse pharmacological approach has the advantage of requiring fewer animal experiments to find new drugs compared to the use of complex compounds which is often time-consuming and cannot guarantee in vivo efficacy.

## 4. Examples of Anthelmintic Drugs from Plants or Plant Extracts

Given the scale of biological diversity of plant species, how is the selection of which ones to examine made? One option is to utilise the rich history of traditional medicinal plant species used by diverse indigenous populations. Over the past 18 years, approximately 215 plant species have been tested against parasitic nematodes such as *Haemonchus contortus*, *Taenina crassicep*, *Ascaris suum*, *Ascaris lumbricoides*, *Taenia solium*, *Schistosoma mansoni* and non-parasitic free-living nematodes like *Caenorhabditis elegans* and *Pheretima posthuman* (Appendix A). These studies have identified considerable anthelmintic activity in vitro and fifteen plants showed anthelmintic efficacy using in vivo models. Plants produce a wide variety of secondary metabolites such as alkaloids, flavonoids, chalcones, coumarins and terpenoids and tannins [3]. In this review, we have highlighted only a few chemical structures of secondary metabolites of many potential compounds based on promising inhibitory concentration (IC 50) values reported in anthelmintic assays in vitro and in vivo (Figure 6). Most of these secondary metabolites are frequently encountered in plants that showed potential anthelmintic properties [3,111,112]. Some of these secondary metabolites showed anthelmintic activity due to their antioxidant properties (carvacrol and thymol) [90], ability to act as neurotoxin (Isoflavonoid-deguelin) [113] and uncoupling of mitochondrial oxidative phosphorylation (polygodial) [114]. It is established that tannin-containing plants/plants extracts have anthelmintic activity against known parasitic nematodes [85], however, further analysis is required to quantify the activity. Moreover, detailed chemical and functional studies are required to identify the active compounds and to understand their mechanisms of action [111].

Even though many plant extracts were validated using in vitro anthelmintic activity, their scientific validation using animal models is limited due to cost of in vivo experiments. Such experiments over the last decade have most commonly used sheep and goats as models [112,115]. Often, successful in vitro results lead to the exploration of the constituents in plant extracts through chemical analysis and subsequent testing of isolated fractions of active compounds. Although this is an innovative approach, it has not yet led to the commercialisation of any anthelmintics.

The best way to determine anthelmintic activity of plant extracts and natural compounds is to test them on naturally infected hosts. However, this requires extensive and sophisticated facilities and large amounts of plant materials. Attempts have been made to test plant extracts against different stages of a parasite life cycle including eggs, larvae [116,117,118] and adults [119] using in vitro identified molecules [85]. However, it can be difficult to obtain large numbers of adult worms for experimentation as this requires the sacrifice of the definitive host and culturing of the adult worms in vitro has also proved extremely challenging.

## 5. Challenges in Natural Product Based Anthelmintic Drug Discovery

Many large pharmaceutical companies have increasingly excluded natural products in drug discovery programs. This has been because of the perceived disadvantages of natural products, which include: (1) difficulties in access and supply, (2) complexities of natural product chemistry, (3) concerns about intellectual property rights, and (4) greater optimism of success with collections of compounds prepared by combinatorial chemistry methods [120,121,122].

Although there has been increased research around the development of anthelmintic drugs, there is a continual need to find new drug agents with new modes of action. Major challenges that exist in the development of anthelminthic drugs include poor understanding of the complicated nature of the life cycles, parasite quantification techniques for effective screening and the ability to develop effective models that represent the complete life cycle of the target parasite [123]. Furthermore, the tendency of variation or specificity of the hosts leads to an argument on the relevance of experimental animals. Although experimental models can mimic many of the biological processes, it remains difficult to attain similar results in infected hosts [123]. A challenge specific to investigating traditional medicines is the availability of medicinal plants which can be restricted to inaccessible areas and require collaboration with indigenous groups. However, through better partnerships, there may be opportunities for indigenous communities to develop new pathways for their economic development through cultivation and harvesting of biologically important plant species.

Presently, most anthelmintic therapy and drug development focus on the parasite adult stages and larvae obtained from the vectors. Some of these key parasitic stages are difficult to obtain in significant numbers and researchers are often compelled to use the more easily available stages which might not reflect the real parasitic stage in the host environment. This results in a poor understanding of the effects of plant-based drugs on the authentic life cycle stages of the parasite [91].

Use of high-throughput screening is the most common method of investigating libraries of drugs. A limitation in anthelmintic drug discovery is the lack of helminth cell lines able to express the receptors or target ion channels needed for high-throughput screening [91]. This has resulted in the use of phenotypic screens, using model worms like *C. elegans* and parasitic worm stages that are maintained in the laboratory environment such as *H. contortus* larval stages in which their viability or behaviour were examined upon exposure to libraries of drugs. However, these screens have various limitations like poor identification of a specific drug target, use of live worms and the use of conventional microscopy for assessment of worm viability [91]. Plant extracts might act on one or more unidentified targets and hence show phenotypic variability with resultant difficulty in characterizing the specific drug target [124]. The use of live worms as an assessment tool in high throughput assays requires an in-depth understanding of their biology. The use of conventional microscopy for assessing worm motility or mortality is often subjective and time-consuming [91]. These methods demonstrate poor sensitivity and specificity as they involve various inherent issues like subjectivity and variations in the available species for the in vitro assay. To facilitate the incorporation of plant extracts in drug development, it is important for the scientific community to address inconsistencies. Currently, there is no system which sufficiently characterizes anthelmintic properties, quantification methods or standardization of plant extracts [91,125]. In addition, there is a dearth of research and understanding of the potentially toxic effects of the plant-based compounds in animals.

When plant extracts are tested for anthelmintic activities, they may contain a high concentration of active compounds. However, this may not correspond to the concentration of active compounds consumed by the host in their diets when they are fed with crude extracts [117]. Furthermore, there is variability in the bioavailability of the plant active compounds in different parts of the gastrointestinal tract of the host [103]. Like commercial anthelmintics that have specific application doses, anthelmintics based on plant extracts will require clear dosage rates. This may be more challenging if there are seasonal or batch-to-batch variations in inactive components. Suggestions have been made to justify the use of plant extracts based on the estimation of overall benefits rather than an evaluation of anthelmintic efficacy *per se.*

Due to these common bottlenecks observed in the extraction of the plant compounds, variations in the screening results, poor specificity and sensitivity, and the requirement for a thorough knowledge of the biology, the screening and development of anthelmintic drugs from plant-based extracts is often considered a high-risk activity for pharmaceutical companies. The minimal benefits and great challenges involved in the screening of plant-based compounds are common factors for poor interest observed by commercial companies. Tapping into the traditional knowledge of the diverse communities/cultures that use plant extracts in their traditional medicine may provide a more beneficial targeted approach compared to high-throughput screening of large synthetic libraries.

## 6. Conclusions

Despite the accepted shortcomings of commercial development of natural products, the vast array of chemically and structurally diverse molecules they exhibit provide an outstanding source of novel compounds for drug discovery [126]. The 2015 Nobel Prize was awarded for the discovery of avermectins and artemisinin, both based on natural products, and the subsequent decline in the incidence of onchocerciasis and malaria raises optimism for natural product-based drug discoveries. There is a long history of medicines derived from natural products, and their impact on human and animal health is undeniable. Above all, natural medicine has been practised over thousands of years in trial and error by a vast number of ethnic groups in geographically distinct areas using diverse plants which gives enormous opportunity to explore new compounds with different modes of action. With the rapid emergence of resistance to available antibiotics and anthelmintics, natural products have the potential to address this problem [127]. Moreover, the extent of nature’s biodiversity is enormous and yet to be fully explored. Taking a multidisciplinary approach by combining natural chemistry and engineered production of variants will extend this biodiversity and help to deliver the next generation of lifesaving anthelmintic drugs and combat the emergence of resistance.

## Figures and Tables

**Figure 1 biomolecules-11-01457-f001:**
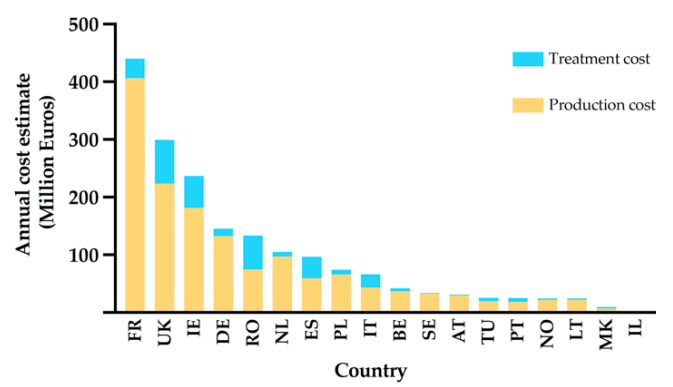
Estimated total annual cost of helminth infections on ruminant livestock production in 18 European and neighbour countries, Annual cost of ruminant industry selected countries estimated at €1.8 billion. Eighty-one percent of this cost is claimed by loss of production and rest due to treatment cost associated with helminth infections. FR = France, UK = United Kingdom of Great Britain and Northern Ireland, IE = Republic of Ireland, DE = Germany, NL = Netherlands, RO = Romania, ES = Spain, PL = Poland, IT = Italy, SE = Sweden, BE = Belgium, NO = Norway, AT = Austria, TU = Tunisia, PT = Portugal, LT = Lithuania, MK = North Macedonia, IL = Israel (Adapted from Charlier et al., 2020 [27]).

**Figure 2 biomolecules-11-01457-f002:**
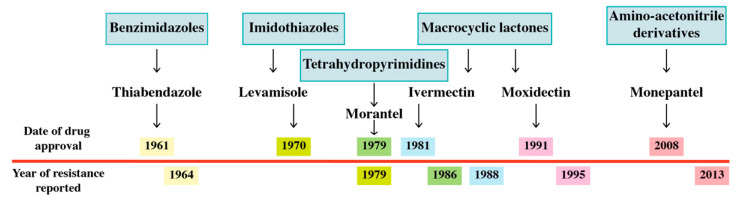
Year of initial drug approval and year of reported resistance in sheep and goats (below line). The individual drugs are colour coded and belong to the major anthelmintic group of drugs currently commercially available. Resistance was reported 3–9 years after the introduction of the drug into the market [38,58,59,60,61,62].

**Figure 3 biomolecules-11-01457-f003:**
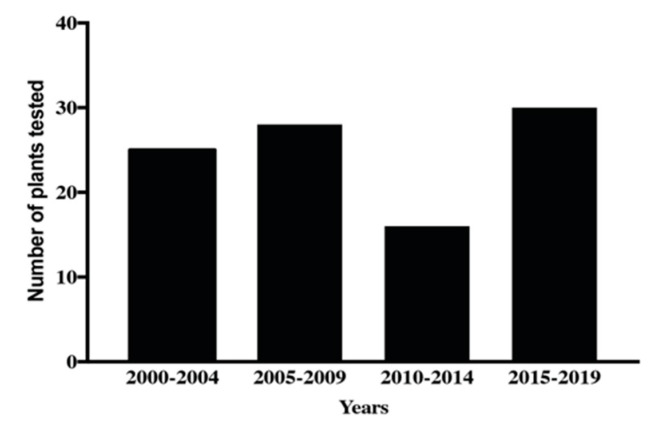
The number of plant extracts tested for anthelmintic activity from 2000 to 2019 according to papers indexed in PubMed, Scopus and Mendeley.

**Figure 4 biomolecules-11-01457-f004:**
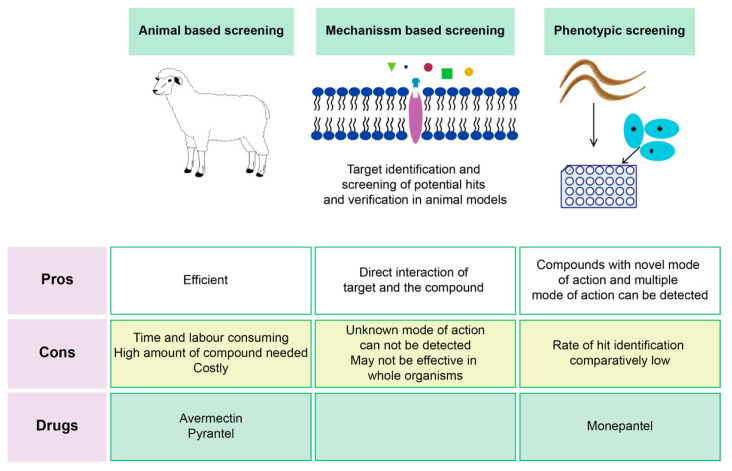
Different primary screening methods for anthelmintic drug discovery with their respective advantages and disadvantages. Phenotypic screening and animal-based screening techniques are forward pharmacological approaches and mechanism-based screening of potential hit compounds is a reverse pharmacological approach.

**Figure 5 biomolecules-11-01457-f005:**
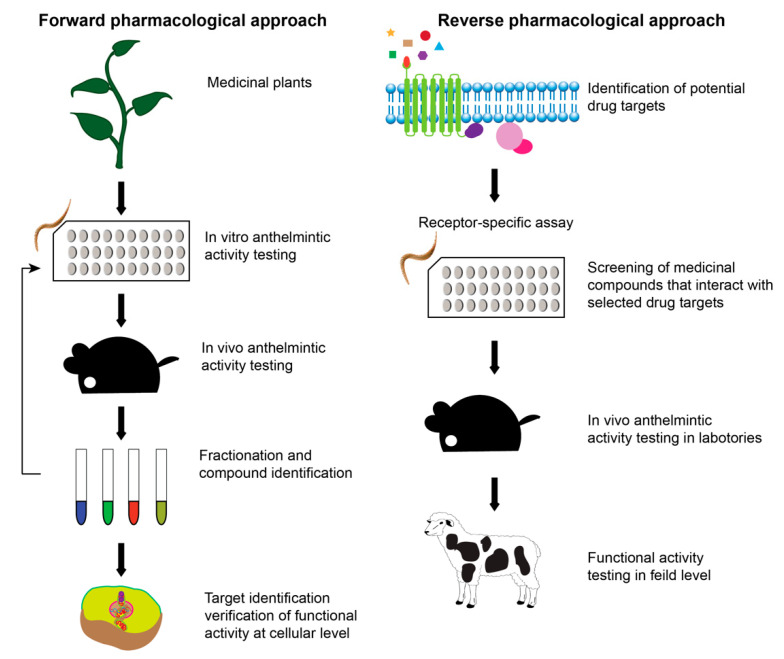
Forward and reverse pharmacological approaches in anthelmintic drug discovery. In forward approaches, potential compounds are first tested on model organisms and changes in the phenotypic/functional activity are monitored. Then, the potential hits are fractionated to identify the lead molecules. These lead molecules are further tested on model parasites and in vivo animal models and the successful candidates are analysed through the target identification process, either by genetic approaches/transcriptomics or metabolomics. In the reverse pharmacological approaches, promising drug targets are first identified and purified and then potential anthelmintic compounds that interact with these targets are screened. The successful compounds are validated in vivo.

**Figure 6 biomolecules-11-01457-f006:**
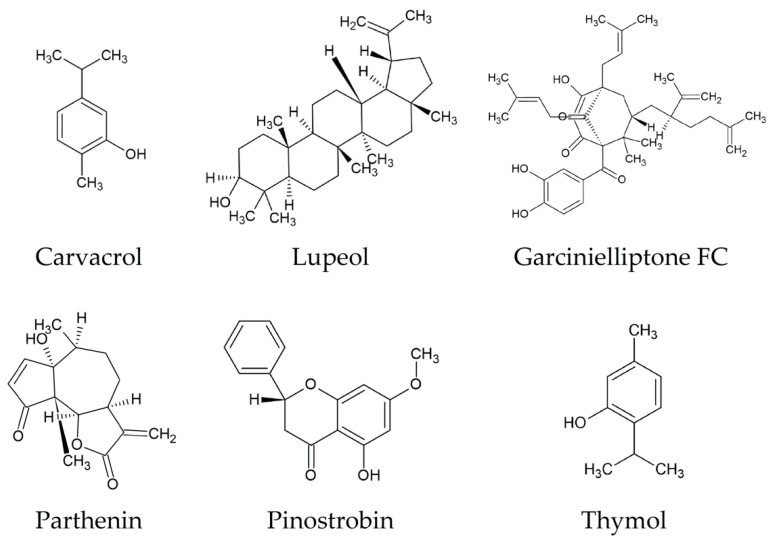
Molecular structures of some secondary metabolites with anthelmintic activity. The structures were generated using ACD/ChemSketch by importing InChI descriptors from PubChem (https://pubchem.ncbi.nlm.nih.gov/, accessed on 28 August 2021).

**Table 1 biomolecules-11-01457-t001:** Estimated prevalence, number of reported deaths and distribution of major parasitic nematode infections in humans.

Disease	Helminth Species	Estimated Prevalence	Reported Deaths	Geographic Distribution	Refs.
**Ascariasis**	*Ascaris lumbricoides*	761.9 million	2700	Worldwide	[2,9]
**Trichuriasis**	*Trichuris trichiura*	1 billion	Not known	Worldwide	[3]
**Hookworm**	*Necator americanus* *Anchylostoma* sp.	428.2 million	Not known	Sub tropics, tropics, and Coastal regions	[2,9]
**Lymphatic filariasis**	*Wuchereria bancrofti,* *Brugia malayi,* *Brugia timori*	119 million	4000	Tropical Africa, Asia, and America	[10,11]
**Strongyloidiasis**	*Strongyloides tercoralis*	30–100 million	Not known	Mainly tropical and sub-tropical regions	[3]
**Loiasis**	*Loa loa*	29.6 million	915	Central Africa	[12]
**Onchocerciasis**	*Onchocerca volvulus*	20.9 million	Not known	Tropical Africa America	[13]
**Trichinosis**	*Trichinella spiralis*	50 million	Not known	Worldwide	[13]
**Enterobiosis**	*Enterobius vermicularis*	>1 billion	Not known	Worldwide	[14,15]
**Dracunculiasis**	*Dracunculiasis medinensis*	>3 million	Not known	Africa and Asia	[3,16]

**Table 2 biomolecules-11-01457-t002:** Target site and mode of action of commercially available anthelmintics.

Class of Anthelmintics	Available Drugs	Target Site	Mode of Action	Refs.
Imidothiazoles	Levamisole	Nicotinic receptor antagonist in body wall muscle	Cause spastic muscle paralysis through prolonged activation of the excitatory nicotinic acetylcholine receptors	[37,39]
Tetrahydro-pyrimidines	Pyrantel, Morantel	Nicotinic receptor antagonist in body wall muscle	Cause spastic muscle paralysis through prolonged activation of the excitatory nicotinic acetylcholine receptors	[37]
Macrocyclic lactones (ML)	Moxidectin, Ivermectin, Avermectin, Milbemycin	Allosteric modulator of GABA—gated chloride ion channels	Inhibit pharyngeal pumping, motility and egg laying	[37,44]
Benzimidazoles	Albendazole, Mebendazole, Febendazole	β-tubulin protein in cytoskeleton	Affect the locomotion and reproduction	[39]
Amino-acetonitrile derivatives (AAD)	Monepantel	Nicotinic acetylcholine receptor subunit	Induce paralysis	[36,45]
Salicylanilides	Closantel, Disophenol	Proton ionophores Bioenergetics	Uncoupling of oxidative phosphorylation	[46]
Organophosphates	Haloxon dichlorvos, Coumaphos Naphthalophos	Acetylcholinesterase	Lead to spastic paralysis	[47]
Spiroindoles	Derquantel	B-subtype nicotinic acetylcholine receptor	B-subtype nicotinic acetylcholine receptor	[47,48]
Clorsulon			Inhibition of phosphoglycerate kinase and mutase	[46]
Praziquantel		Ca ion channels	Increase in membrane permeability towards calcium results in increase calcium influx leading to muscular contracture	[49]
Diamphenethide			Inhibition of malate metabolism	[50]
Piperazine	Piperazine	GABA receptors	Binding to muscle membrane GABA receptors causing hyperpolarization of nerve endings	[51]

**Table 3 biomolecules-11-01457-t003:** Mechanism of anthelmintic resistance for the commonly used anthelmintics in humans and livestock species.

Anthelmintic Drugs	Helminth Species	Reported Mechanism of Drug Resistance	Refs
Oxamniquine	Schistosomes	1. Deficiency in drug-activating enzyme—sulfotransferase 2. Loss of function in drug activating enzyme sulfotransferase	[68,69]
Praziquantel	Schistosomes	1. Deficiency in drug-activating enzyme—sulfotransferase	[57,70]
Macrocyclic lactones (Ivermectin and Avermectin)	*Haemonchus contortus*	1. Alter the structure of GluCl channel subunits or GABA receptor 2. Overexpression of P-glycoproteins	[71,72,73]
Imidazothiazoles (Levamisole)	Trichostrongylids	1. Reduction in the number of levamisole receptors in resistant trichostrongylids 2. Reduction in the affinity of levamisole receptors for levamisole	[74]
Closantel	*Fasciola* spp. *Haemonchus contortus*	1. Reduction of feeding by resistant worms 2. Reduction of dissociation of the drug-albumin complex in the worm gut 2. Overexpress the drug target (P-glycoprotein mediated increased drug efflux)	[75]
Amino-acetonitrile derivatives (Monepantel)	*Haemonchus contortus*	1. Mutations in the drug target gene (*Hco-mptl-1*)	[67]

## Data Availability

Not applicable.

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
