# Peer review of "Natural Products Are a Promising Source for Anthelmintic Drug Discovery"

_biomolecules, 2021, doi:10.3390/biom11101457_

Round 1
Reviewer 1 Report
The work entitled “Natural products are a promising source of anthelmintic drug discovery” proves to be insightful and well developed. However, several bibliographic data refer to generic and not recently published articles. Authors could consider to add in the bibliography some most up to date specific examples regarding the potential anthelmintic activity of plants, such as the below incuded suggestions:
- Ndlela, S.Z.; Mkwanazi, M.V.; Chimonyo, M. In vitro efficacy of plant extracts against gastrointestinal nematodes in goats. Trop Anim Health Prod. 2021;53(2):295. Published 2021 Apr 29. doi:10.1007/s11250-021-02732-0
- Castagna, F.; Britti, D.; Oliverio, M.; Bosco, A.; Bonacci, S.; Iriti, G.; Ragusa, M.; Musolino, V.; Rinaldi, L.; Palma, E.; Musella, V. In Vitro Anthelminthic Efficacy of Aqueous Pomegranate (Punica granatum) Extracts against Gastrointestinal Nematodes of Sheep. Pathogens. 2020 Dec 18;9(12):1063. doi: 10.3390/pathogens9121063. PMID: 33353177; PMCID: PMC7766728.
- Soren, A.D.; Chen, R.P.; Yadav, A.K. In vitro and in vivo anthelmintic study of Sesbania sesbanbicolor, a traditionally used medicinal plant of Santhal tribe in Assam, India. J Parasit Dis. 2021 Mar;45(1):1-9. doi: 10.1007/s12639-020-01267-9. Epub 2020 Sep 9. PMID: 33746380; PMCID: PMC7921245.
- Badar, S.N.; Iqbal, Z.; Sajid, M.S.; Rizwan, H.M.; Shareef, M.; Malik, M.A.; Khan, M.N. Comparative anthelmintic efficacy of Arundo donax, Areca catechu, and Ferula assa-foetida against Haemonchus contortus. Rev Bras Parasitol Vet. 2021 May 28;30(2):e001221. doi: 10.1590/S1984-29612021028. PMID: 34076046.
In addition, Authors should check the consistency of bibliographic references. An example of it, but not limited to that, is the journal Veterinary parasitology reference is cited differently in references 18 and 25 and Trends in parasitology in references 17 and 23.
Author Response
Authors could consider to add in the bibliography some most up to date specific examples regarding the potential anthelmintic activity of plants, such as the below included suggestions:
Response:
We have updated the references to include more up to date examples.
Authors should check the consistency of bibliographic references
Response:
We have checked the references for consistency of journal names
Reviewer 2 Report
In this review the authors highlight how natural products can be used as potential Anthelmintics, the rationale behind using natural products, current stage of research and future directions. Although the review tries to touch upon a lot of those aspects, the review could become even more informative if some of these additional items are addressed:
• In lines 45 -59 although the authors say there is no clear data about the case numbers. It would be a good idea to show some graphics indicating which region/ countries are most affected and an association of what types of infections and diseases are most prevalent because of these helminthic infections.
• In lines 62-65 it will be good to show some figures about the economic loss/ impact that is caused, please find some reports indicating that.
• When the authors discuss the current anthelmintics they must highlight how there is an issue with off target effects and toxicity (PMID: 25189803, PMID: 23036678). The authors also need to go in depth about the mode of action/ pharmacology of these compounds, chemical structures/ scaffolds, targets etc. Also tabulating this information will be a good idea. Refer to http://www.wormbook.org/chapters/www_anthelminticdrugs/anthelminticdrugs.html for more information.
• Show some graphics or a table where you can give information about what stages of the lifecycle do these drugs act on.
• Another table that would be good to have, are the list of key mutations leading to drug-resistance.
• Although the authors mention how natural compounds are being used, they need to show what specific natural compounds are used extensively or have shown some potential compounds (Thymol, Caravacrol, Parthenin etc.), their pharmacology (If they are Neurotoxins, Antifeedants etc.) and chemistry.
• When the authors talk about the screening methods, they should highlight what the major targets that have been studied and utilized extensively for drug discovery purposes (Pros and cons) and potential targets.
• Also, the authors should provide more information about the types of screens (phenotypic and enzymatic), types of compounds/ libraries (pure compounds, mixtures, plant extracts) that are being used or prioritized, the models that are being used etc.
• Also, one of the key items that can be highlighted is using cheminformatic tools and natural compound databases such as SAR, GNPS, Chem-GPS etc. which can help with identifying and fishing out compounds accelerating the drug discovery process.
• Minor comment: The image resolutions can be improved.
Author Response
1/ It would be a good idea to show some graphics indicating which region/ countries are most affected and an association of what types of infections and diseases are most prevalent because of these helminthic infections.
Response:
We have added a new table (Table 1) and discussion in the text about the geographic distribution of key helminthic infections.
2/ ….it will be good to show some figures about the economic loss/ impact…
Response:
We have added a brief discussion of the costs of gastrointestinal nematodes (GIN) in the text and a new figure (Figure 1) that outlines the economic costs of helminth infections in the context of production and treatment cost in European countries as an example.
3/ When the authors discuss the current anthelmintics they must highlight how there is an issue with off target effects and toxicity (PMID: 25189803, PMID: 23036678). The authors also need to go in depth about the mode of action/ pharmacology of these compounds, chemical structures/ scaffolds, targets etc. Also tabulating this information will be a good idea. Refer to http://www.wormbook.org/chapters/www_anthelminticdrugs/anthelminticdrugs.html for more information. Line 99-101
Response:
We have included a brief discussion and included a new table (Table 2) that indicates the mode of action of specific anthelmintics. This review is about the discovery of new anthelmintics and we acknowledged in the text that toxicity is a key issue for generating the next generation of drugs. We feel a discussion about the toxicity of existing anthelmintics is outside the scope of this review.
4/ Show some graphics or a table where you can give information about what stages of the lifecycle do these drugs act on.
Response:
As this review is forward-looking at how plant extracts can be a new source of anthelmintics, we feel a table giving information about lifecycle stage susceptible to existing drug treatments is outside the scope of this article.
5/ Another table that would be good to have, are the list of key mutations leading to drug-resistance
Response:
We have added a table (Table 3) that outline the key anthelmintics resistance mutations.
6/ “…they need to show what specific natural compounds are used extensively or have shown some potential compounds (Thymol, Caravacrol, Parthenin etc.), their pharmacology (If they are Neurotoxins, Antifeedants etc.) and chemistry.
Response:
We have a figure (Figure 6) that shows the chemical structures of selected compounds.
7/When the authors talk about the screening methods, they should highlight what the major targets that have been studied and utilized extensively for drug discovery purposes (Pros and cons) (line 207 & figure) and potential targets.
Response:
Figure 4 contains a description of the Pros and Cons of the different discovery approached. The text also includes a discussion of the advantages and challenges of the various screen approaches (sections 3 and 4).
8/ Also, the authors should provide more information about the types of screens (phenotypic and enzymatic), types of compounds/ libraries (pure compounds, mixtures, plant extracts) that are being used or prioritized, the models that are being used etc.
Response:
Section 3.1 “Screening techniques of plant-based compounds” address the phenotypic and enzymatic approaches and the type of compounds that can be screened. The focus of this review is on the advantages of plant extracts as a new source of anthelmintics compounds compared to existing compounds/ libraries. Therefore, we do not believe including more information about compounds/libraries fits the theme of the review.
9/ Also, one of the key items that can be highlighted is using cheminformatic tools and natural compound databases such as SAR, GNPS, Chem-GPS etc. which can help with identifying and fishing out compounds accelerating the drug discovery process.
Response:
The focus of this review is to discover new plant-derived compounds for helminth control, not repurposing existing compounds, therefore we do not think it is appropriate to include a section examining the use of cheminformatic tools/databases.
Reviewer 3 Report
The paper submitted by Dilrukshi Jayawardene et co-workers is a review on the role of natural products as anthelmintic compounds. The paper is well written and can be a useful source in the future for other researchers. However, before it can be considered for publication in Biomolecules some corrections and improvements are needed.
1) the authors should put more emphasis on plant materials used or investigated as anthelmintic drugs - a full review of previous literature, table with list of plants investigated should be added; the authors should discuss previous research in more comprehensive manner not only examples of plant materials
2) please discuss which groups of natural products (pure compounds) were considered as promising anthelmintic agents - chemical structures should also be presented
Author Response
1/ The authors should put more emphasis on plant materials used or investigated as anthelmintic drugs - a full review of previous literature, table with a list of plants investigated should be added; the authors should discuss previous research in a more comprehensive manner not only examples of plant materials
Response:
We have included an extensive table shown plant materials examined for anthelmintic activity (Supplemental Table 1).
2/ Please discuss which groups of natural products (pure compounds) were considered as promising anthelmintic agents - chemical structures should also be presented
Response:
We have added a brief discussion n promising anthelmintic agents and added a table with the chemical structures of selected natural products.
Round 2
Reviewer 3 Report
No further comments.